# Serum TNF -α, IL-10 and IL-2 Trajectories and Outcomes in NSCLC and Melanoma Under Anti-PD-1 Therapy: Longitudinal Real-World Evidence from a Single Center

**DOI:** 10.3390/cimb47090746

**Published:** 2025-09-11

**Authors:** Alina Miruna Grecea-Balaj, Olga Soritau, Ioana Brie, Maria Perde-Schrepler, Piroska Virag, Eva Fischer-Fodor, Nicolae Todor, Mihai Cenariu, Ioana Nedelea, Tudor Eliade Ciuleanu

**Affiliations:** 1Department of Medical Oncology, Oncology Institute “Prof. Dr. Ion Chiricuta”, 400015 Cluj-Napoca, Romania; miruna_grc@yahoo.com; 2Department of Medicine, University of Medicine and Pharmacy “Iuliu Hatieganu”, 400012 Cluj-Napoca, Romania; 3Department of Tumor Biology and Radiobiology, Oncology Institute “Prof. Dr. Ion Chiricuta”, 400015 Cluj-Napoca, Romania; olgasoritau@yahoo.com (O.S.); ioanabrie@yahoo.com (I.B.); pmariaida@yahoo.com (M.P.-S.); virag.piroska@yahoo.com (P.V.); eficherfodor@yahoo.com (E.F.-F.); 4Department of Biostatistics, Oncology Institute “Prof. Dr. Ion Chiricuta”, 400015 Cluj-Napoca, Romania; tdrnicolae@yahoo.com; 5Department of Animal Reproduction, University of Agricultural Sciences and Veterinary Medicine, 400372 Cluj-Napoca, Romania; mihai.cenariu@usamvcluj.ro; 6Department of Medical Oncology, Emergency County Hospital “Dr. Constantin Opris”, 430031 Baia Mare, Romania; ioanaiurca@yahoo.com

**Keywords:** immune checkpoint inhibitor, NSCLC, melanoma, cytokines, Il-2, IL-10, TNF-α

## Abstract

This prospective single-center study examined associations between serum cytokines—TNF-α, IL-2, and IL-10—and outcomes in stage IV non-small cell lung cancer (NSCLC, *n* = 43) and melanoma (*n* = 15) patients treated with Nivolumab at the Oncology Institute in Cluj-Napoca, Romania. Cytokines were measured at baseline (NSCLC: *n* = 43; melanoma: *n* = 15), 3 months (NSCLC: *n* = 20; melanoma: *n* = 7), and 6 months (NSCLC: *n* = 10; melanoma: *n* = 5). Melanoma patients showed sustained IL-2 and TNF-α increases, while NSCLC patients displayed heterogeneous cytokine dynamics. In NSCLC, elevated IL-10 at 3 months correlated with shorter survival (ρ = −0.51, 95% CI −0.78 to −0.12, *p* = 0.022) and poorer response (ρ = −0.65, 95% CI −0.86 to −0.23, *p* = 0.002). TNF-α showed a borderline association with response (ρ = −0.44, 95% CI −0.74 to 0.01, *p* = 0.050). In melanoma, 3-month TNF-α was inversely associated with survival (ρ = −0.82, 95% CI −0.97 to −0.15, *p* = 0.023) and response (ρ = −0.90, 95% CI −0.99 to −0.39, *p* = 0.006). Strong inter-cytokine correlations were observed (NSCLC: TNF-α vs. IL-10, ρ = 0.60, 95% CI 0.19–0.82; melanoma: ρ = 0.93, 95% CI 0.44–0.99). Baseline cytokines had limited utility, particularly in melanoma due to the small sample size. The most informative finding was the association of elevated 3-month IL-10 with adverse outcomes in NSCLC. These results support the value of dynamic cytokine monitoring in immunotherapy and warrant validation in larger cohorts.

## 1. Introduction

The advent of immune checkpoint inhibitors (ICIs), particularly those targeting the PD-1/PD-L1 axis, has revolutionized the management of various malignancies, including non-small cell lung cancer (NSCLC) and melanoma. Despite these advances, a substantial proportion of patients either fail to respond or experience immune-related adverse events (irAEs), limiting the broad applicability of ICIs and highlighting the pressing need for reliable predictive biomarkers to guide patient stratification and therapeutic decision-making [1,2,3].

Among emerging candidates, cytokines—soluble signaling proteins secreted by immune and tumor cells—have emerged as promising biomarkers due to their central role in modulating immune responses, their detectability in peripheral blood, and their suitability for dynamic monitoring [4,5].

Several studies have demonstrated that distinct cytokine profiles may serve as early indicators of treatment efficacy or toxicity. Elevated levels of cytokines such as TNF-α, IFN-γ, IL-6, IL-8, and TGF-β have been linked to both favorable responses and increased risk of irAEs in patients receiving ICIs [6,7]. In contrast, high levels of immunosuppressive cytokines like IL-10 (and certain pro-tumor chemokines such as IL-8, which can recruit suppressive myeloid cells) are associated with poorer outcomes, potentially by promoting T-cell exhaustion and tumor immune evasion [3].

Empirical evidence further supports these patterns: increased baseline IL-2 has correlated with better progression-free and overall survival under ICI therapy [8,9], whereas elevated chemokines like MIG (CXCL9), eotaxin, and IP-10 (CXCL10) have been tied to a greater risk of developing irAEs [4,6,10]. High IL-8 is consistently identified as a negative prognostic indicator across multiple cancers, reinforcing its role in an immunosuppressive tumor microenvironment [3]. Even prototypical inflammatory cytokines such as TNF-α and TGF-β have been implicated as potential predictors of irAEs, highlighting that both the magnitude of immune activation and the balance of regulatory signals can influence toxicity [2,4].

These three cytokines each play a pivotal role: TNF-α is a major inflammatory mediator that can enhance T-cell responses (but also contribute to tissue inflammation/toxicity), IL-2 is a critical T-cell growth factor underpinning effector lymphocyte expansion, and IL-10 is a prototypical anti-inflammatory cytokine that dampens immune responses and can drive T-cell exhaustion. All have well-documented involvement in cancer immunology and ICI activity, making them logical candidates for longitudinal monitoring [2,4,9,11]. Moreover, they are readily measurable with standard enzyme-linked immunosorbent assay (ELISA) techniques, which facilitates feasible serial testing in a clinical setting. While other cytokines such as IL-6 and IFN-γ also exhibit prognostic potential in immunotherapy [12,13], we prioritized a concise panel of TNF-α, IL-2, and IL-10 to balance biological relevance with practical assay feasibility for real-world, repeated measurements.

Advanced cytokine profiling techniques, including multiplex assays and machine learning approaches [14], have further refined the identification of cytokine patterns predictive of therapeutic outcomes and adverse events, paving the way for more personalized immunotherapy strategies [7,15,16]. Moreover, given their ability to disrupt tumor cell membranes and trigger robust immune responses, oncolytic peptides are being explored in combination with established immunotherapies and as part of personalized treatment regimens, further expanding the toolkit for precision oncology [17,18,19].

As research progresses, integrating cytokine profiling into clinical practice holds the potential to optimize immunotherapy by identifying patients most likely to benefit, minimizing toxicity, and unraveling the underlying mechanisms of immune modulation in cancer [20,21]. By bridging preclinical insights with clinical validation, this work aims to advance cytokine-based biomarkers from exploratory tools to actionable guides in precision oncology.

This study seeks to address these gaps by providing real-world longitudinal data on serum TNF-α, IL-2, and IL-10 dynamics in a cohort of patients with stage IV NSCLC and melanoma undergoing anti-PD-1 therapy with Nivolumab.

Cytokine concentrations were measured at baseline and again at 3 and 6 months and correlated with clinical outcomes to evaluate their prognostic and predictive utility in the context of immunotherapy response. Notably, no single primary endpoint or hypothesis was defined a priori; instead, our objective was to broadly assess whether baseline values or temporal changes in these cytokines could serve as biomarkers of treatment outcomes. By including two distinct tumor types, NSCLC and melanoma, we also explored whether cytokine-outcome associations are consistent across different immunogenic tumor environments or vary by cancer type. Unlike many previous reports that relied on retrospective data or pooled analyses, this prospective study integrates dynamic biomarker measurements with observed patient survival and response outcomes in a defined cohort. Such an approach moves beyond speculative correlations in the literature toward more tangible clinical correlates, aiming to advance cytokine profiling from an exploratory research tool to an actionable guide in precision oncology.

## 2. Materials and Methods

### 2.1. Study Design

This is a prospective observational single-center clinical study that was conducted at the Oncology Institute “Prof. Dr. I Chiricuta” in Cluj-Napoca, Romania, between December 2017 and May 2019. The cohort comprised 58 patients diagnosed with stage IV non-small cell lung cancer (NSCLC) and stage IV melanoma who underwent immunotherapy with anti-PD-1 Nivolumab at the institution as part of standard-of-care treatment. Serum concentrations of tumor necrosis factor-alpha (TNF-α), interleukin-2 (IL-2), and interleukin-10 (IL-10) were longitudinally assessed using standardized assays at three time points: baseline (pre-treatment), 3 months post-treatment initiation, and 6 months post-treatment initiation, this being the only study-specific procedure beyond routine care—Figure 1. The analysis aimed to evaluate temporal changes in cytokine profiles associated with immunotherapy response in advanced-stage malignancies. Given the immunogenic differences between NSCLC and melanoma, data were analyzed both as a combined cohort and stratified by tumor type to preserve biological specificity. Comparative analyses were conducted for each cancer type to highlight differential cytokine trends.

### 2.2. Inclusion and Exclusion Criteria

Participants were required to have a pathology-confirmed diagnosis of either stage IV melanoma or non-small cell lung cancer. Eligible patients were aged 18 years or older, had not previously received immunotherapy, and were either in first-line or subsequent lines of therapy. Written informed consent was obtained from all participants. Exclusion criteria included pregnancy or lactation, any prior history of malignancy (apart from non-melanoma skin cancer or in situ cervical carcinoma), and any previous exposure to immunotherapeutic agents.

### 2.3. Methods and Statistical Analysis

Analyses were conducted in a descriptive and exploratory manner, without predefined hypotheses. Results were summarized in tables and interpreted descriptively. Categorical variables were reported as counts and percentages, while continuous variables were described using standard statistics (number of observations, missing values, mean, standard deviation, median, minimum, and maximum) [22].

Statistical analyses were conducted using IBM SPSS Statistics software version 30, with a significance threshold set at *p* < 0.05 and no multiple-testing correction applied due to the exploratory nature and limited sample size of the study.

Longitudinal biomarker analysis was performed using linear mixed models with IBM SPSS statistics. Correlations analyses were performed employing the non-parametric Spearman rank correlation test. Spearman’s rank correlation coefficient was used to assess associations between cytokine parameters and clinical outcomes (e.g., survival). This method was chosen for its robustness to non-normal distributions and outliers, which are common in cytokine serum level datasets. Fixed effects included time (categorical), tumor type (NSCLC vs. melanoma), and their interaction. Residual within-subject correlation was modeled as AR (1). Parameters were estimated by REML; ML was used when comparing covariance structures. We report model-based means (estimated marginal means) and contrasts with 95% confidence intervals (CIs) and exact *p*-values.

As a more relevant approach, we used Cox proportional hazards models adjusted for PD-L1, smoking, ECOG PS (0–2), line of nivolumab (1–4), corticosteroid and antibiotics use, tumor type (NSCLC vs. melanoma). Cytokines (TNF-α, IL-2, IL-10) were modeled at baseline, 3 months, and 6 months.

To capture longitudinal dynamics, we used a counting-process (start–stop) Cox model with intervals 0–3, 3–6, and >6 months. Cytokines were treated as time-varying covariates using last-observation-carried-forward across intervals. We applied log1p transformation and z-scaling to cytokines. Hazard ratios (HRs) are reported per 1 SD increase in log1p (cytokine). We used cluster-robust standard errors at the patient level. Melanoma-only sensitivity models additionally adjusted for BRAF.

A single primary analysis was pre-specified (NSCLC 3-month IL-10 vs. overall survival). All other analyses were considered exploratory. To control type-I error in the presence of multiple testing, we applied Benjamini–Hochberg false discovery rate (FDR) adjustment within families of related tests: for Spearman correlation matrices (defined by tumor type × timepoint) and across cytokines in the time-updated Cox joint model. We report both unadjusted *p* and FDR-adjusted q values.

Spaghetti plots depicting individual patient cytokine trajectories over time were generated using Python (v3.9) with the matplotlib and seaborn libraries. Serum concentrations of IL-2, IL-10, and TNF-α (pg/mL) were plotted on a logarithmic y-axis to accommodate the wide range of cytokine levels observed. For each cytokine, line plots were created with timepoints on the x-axis (baseline, 3 months, 6 months) and cytokine concentrations on the y-axis. Each line represents a single patient’s longitudinal profile, with unique colors assigned to distinguish individual trajectories. Separate plots were generated for the NSCLC and melanoma cohorts. The figure layout was to display the three cytokines side-by-side, facilitating comparison across conditions.

Heatmaps were generated using the Seaborn data visualization library in Python with the heatmap () function used to display Spearman correlation coefficients between cytokines. Color intensities represent correlation strength, and numerical values within each cell denote correlation coefficients. Seaborn builds on the Matplotlib library and is optimized for statistical graphics in Python.

Cytokine concentrations were determined utilizing enzyme-linked immunosorbent assay (ELISA) DuoSet kits (R&D Systems). ELISA assays were conducted per manufacturer protocols. Assays had intra-assay CVs < 10% and inter-assay CVs < 15%. Sensitivity thresholds were as follows: TNF-α (4.4 pg/mL), IL-10 (3.9 pg/mL), and IL-2 (5.1 pg/mL). All samples were run in duplicate with internal positive and negative controls included per plate to ensure reproducibility and quality control.

Any cytokine measurement that fell below the assay’s LOD was considered non-quantifiable and excluded from our analyses. We did not substitute or impute values for these non-detectable readings, they were treated as missing data and omitted from statistical analysis.

All patients followed strict preanalytical protocols. They were required to fast for at least 8 h (overnight) before blood draw, and all blood samples were collected in the morning, between 8:00 and 10:00 AM. By synchronizing collection times to the morning after an overnight fast, we minimized variability due to circadian rhythms and postprandial changes in cytokine levels. Immediately after collection and processing, samples were frozen at −80 °C to preserve cytokine stability. Maintaining a short storage period at ultra-low temperature prevented degradation of the cytokines or loss of signal over time.

To minimize inter-assay and inter-plate variability, all longitudinal samples from the same patient were run on the same ELISA plate in a single batch. Running all timepoints for each patient on one plate enhances the comparability of the readings by eliminating potential inter-plate differences in assay conditions.

Given the high attrition at 3 and 6 months (53.4% and 74.1%, respectively), no imputation was performed. Missing data were assumed to be missing at random (MAR). Sensitivity analyses comparing baseline characteristics of patients with and without follow-up cytokine data revealed no significant differences, mitigating concerns about selection bias.

### 2.4. Clinical Endpoints

Treatment response was assessed using RECIST 1.1 criteria via imaging at standard intervals (every 12 weeks). Responses were categorized as complete response (CR), partial response (PR), stable disease (SD), or progressive disease (PD). Overall survival (OS) was defined as time from initiation of Nivolumab to death or last follow-up. Best response was defined as the best radiologic outcome achieved during therapy.

### 2.5. Study Purpose

This study investigates the integration of cytokine profiling into clinical practice to enhance therapeutic optimization in immunotherapy-responsive malignancies, with a focus on elucidating mechanistic insights into immune microenvironment modulation in NSCLC and metastatic melanoma. The methodology prioritizes longitudinal cytokine monitoring to dynamically assess treatment efficacy and immune reconstitution patterns, particularly in cancers characterized by high tumor mutational burden and checkpoint inhibitor susceptibility such as NSCLC and melanoma.

## 3. Results

### 3.1. Patients’ Characteristics

Following a cutoff date for cytokine monitoring as of December 2019, statistical analysis was conducted in May 2025 using IBM SPSS software (Version 30.0), revealing a median follow-up duration of 22 months. A subsequent June 2025 updated survival analysis revealed exceptional responders among patients with extended survival. Comprehensive baseline cytokine profiles were obtained for all participants. Follow-up cytokines were available for 27/58 (46.6%) at 3 months and 15/58 (25.9%) at 6 months (Supplement) due to disease progression or death. We provide baseline comparisons for those with vs. without 3/6-month cytokines (Supplement). In mixed-effects models, missingness was treated as MAR (missing at random) but may be informative given progression or death. We therefore report sensitivity analyses (pattern-mixture with dropout-by-time term) and emphasize the time-updated Cox models, in which participants contribute person-time until the event. All cytokine measurements were obtained during Nivolumab monotherapy, and no concurrent systemic therapies were administered. This attrition pattern highlights challenges in maintaining longitudinal biomarker assessments in advanced cancer populations.

Out of 58 patients, the vast majority were males (67.2%) with a median age of 62.5 years (range: 35–83), where lung cancer represented 77.6% of cases and melanoma 22.4%. Smoking history was prevalent, with 70.7% identified as current or former smokers. The most frequent histology in NSCLC patients was adenocarcinoma (58.1%) and PD-L1 status was equally distributed in the NSCLC cohort. 36.2% of patients [21] had an active use of steroids at baseline defined by a time ranging from 14 days before starting ICI up to 30 days after treatment initiation. Treatment timelines showed a median overall survival of 21.1 months between Nivolumab initiation and last follow-up or death, with the median number of administered cycles being 17. Nivolumab was typically administered as a second line, though utilization ranged from first- to fourth-line treatments. 50% of patients underwent subsequent lines of therapy, from chemotherapy to targeted treatment and combinations. Most patients had either progression (48.3%) or stable disease (43.1%) as best response and only 6.9% had a complete response. Longitudinal data revealed extended follow-up periods, with some patients remaining under observation until June 2025, indicating ongoing survival tracking in this real-world dataset—Table 1.

### 3.2. Cytokines Timeline Trends in Both Cohorts

#### 3.2.1. IL-2 Response Patterns

NSCLC patients show heterogeneous IL-2 responses with two distinct patterns: some patients begin with elevated baseline levels (75–80 pg/mL) that decrease over the treatment period, while others maintain consistently low levels throughout (10–20 pg/mL). This variation suggests differing baseline immune activation states and variable responsiveness to PD-1 blockade. In contrast, melanoma patients show more pronounced and sustained IL-2 increases, with several individuals (e.g., patient 2MM) reaching levels exceeding 300 pg/mL by six months. The observed increase in IL-2 levels among melanoma patients treated with Nivolumab is likely attributable to activation and expansion of CD4^+^ helper T cells and CD8^+^ cytotoxic T lymphocytes, which are principal producers of IL-2 following T-cell receptor engagement. This pattern suggests stronger T-cell activation and proliferation in response to Nivolumab in melanoma compared to NSCLC.

#### 3.2.2. IL-10 Dynamics

NSCLC patients maintain low IL-10 levels, with one notable exception showing a dramatic spike at 3 months (~280 pg/mL). This isolated elevation may reflect a short-lived regulatory immune response, though no immune-related adverse events were associated. Melanoma patients exhibit more variable IL-10 patterns, with some patients showing inflated baseline levels that fluctuate during treatment. This variability suggests diverse regulatory immune mechanisms at play within the melanoma group, potentially reflecting differences in immunosuppressive cell populations or feedback responses to T-cell activation.

#### 3.2.3. TNF-α Inflammatory Response

TNF-α trends differ across cohorts. In melanoma patients, several individuals demonstrate consistent increases over time, with levels reaching over 1000 pg/mL in some cases (e.g., 2MM, 15MM) by 6 months. This pattern suggests a trend towards pro-inflammatory response, potentially driven by increased immune system engagement. In contrast, NSCLC patients show more variable TNF-α trajectories. While some exhibit rising or falling levels, most remain near baseline, reflecting a less uniform or muted inflammatory response to therapy—Figure 2.

Cytokine concentrations (pg/mL) are plotted on a logarithmic scale to accommodate the wide dynamic range of values. Measurements were taken at three timepoints: baseline (prior to immunotherapy, timepoint 0), 3 months (timepoint 1), and 6 months (timepoint 2) after treatment initiation. Each line represents the cytokine profile of an individual patient across the treatment course, illustrating inter-individual variability in immune responses within and between cohorts.

This data suggests melanoma patients respond more robustly to Nivolumab with stronger pro-inflammatory cytokine activation (IL-2, TNF-α). These patterns align with known biological differences in PD-1 inhibitor responses between epithelial tumors (NSCLC) and immunogenic melanomas. The heterogenous NSCLC responses may reflect the tumor’s more complex immune microenvironment and variable PD-L1 expression patterns. These differential cytokine kinetics show distinct temporal patterns between the two cohorts—Figure 3.

Box-and-whisker plots illustrate longitudinal changes in cytokine concentrations measured at baseline (pre-treatment), 3 months, and 6 months. Data from both NSCLC and melanoma patients are shown to highlight inter-cohort variability. The boxes represent the interquartile range (IQR), horizontal lines mark the median, whiskers extend to the most extreme values within 1.5× IQR, and individual dots denote outliers. Distinct patterns of immune activation and regulation are evident between cancer types, with melanoma patients generally exhibiting stronger pro-inflammatory responses (e.g., IL-2 and TNF-α elevations), while IL-10 dynamics suggest differential regulatory feedback.

Longitudinal biomarker analysis was performed using linear mixed models with IBM SPSS statistics.

The two cancer cohorts uncover unique cytokine profiles, exhibiting significant variability between patients. The linear mixed-effects analysis examined TNF-α levels over time (baseline, 3 months, 6 months) across the two cancer types (melanoma vs. NSCLC) and included random intercepts for each participant and accounted for autocorrelation with an AR (1) residual structure. Considering the time effect, there was an initial increase; specifically, TNF-α rose from baseline (168.6 pg/mL, SE = 34.9) to 3 months (242.6 pg/mL, SE = 41.0; *p* = 0.054), then leveled off by 6 months (244.3 pg/mL, SE = 49.7). Melanoma patients exhibited a marginally higher overall TNF-α level compared to NSCLC patients, F (1, 70.3) = 3.81, *p* = 0.055. The interaction between time and cancer type was not significant, F (2, 47.1) = 0.99, *p* = 0.381, indicating that the pattern of TNF-α changes over time was similar for both cancer groups. Regarding IL-10 levels, they show a significant rise by the 3-month mark and maintain elevated concentrations through 6 months of monitoring. While cancer type did not significantly affect IL-10 concentrations, melanoma cases exhibited a more pronounced upward trajectory in early measurements. As for IL-2 it simply did not show any consistent change or group differences over the six-months period—Appendix A.

However, in time-updated covariate-adjusted joint models (TNF-α + IL-2 + IL-10 entered together), IL-2 was significantly associated with lower hazard (HR 0.53 per SD, 95% CI 0.36–0.78, *p* = 0.0013, q = 0.0078) whereas TNF-α was associated with higher hazard (HR 4.41 per SD, 95% CI 1.15–16.98, *p* = 0.031, q = 0.093). IL-10 was not significant as stated before. Corresponding static (single-timepoint) adjusted models, single-cytokine and joint, showed no significant associations. The strongest static signal (baseline TNF-α) was borderline (*p*≈0.05) before transformation and did not remain significant after transformation—Appendix A.

### 3.3. Baseline Cytokine Levels as Predictors of Treatment Response and Survival

In the NSCLC cohort, the analysis of the 43 patients revealed several key correlations between baseline cytokine levels and clinical outcomes. Baseline TNF-α displays a moderate positive correlation with IL-2 (ρ = 0.544, *p* < 0.001) and a weaker but significant positive correlation with IL-10 (ρ = 0.386, *p* = 0.011), suggesting coordinated inflammatory signaling. Conversely, TNF-α showed significant negative correlations with both survival duration (ρ = −0.358, *p* = 0.018) and treatment response (ρ = −0.364, *p* = 0.017), indicating higher baseline TNF-α levels were associated with poorer clinical outcomes. IL-2 and IL-10 exhibit no significant correlations with outcomes (*p* > 0.05), limiting their prognostic utility in this cohort.

Within NSCLC subtypes, we also assessed Spearman correlations between overall survival and cytokines at baseline, 3 and 6 months. In adenocarcinoma, no correlation reached statistical significance after FDR control (strongest unadjusted signal: IL-10 at baseline, ρ = −0.34, *p* = 0.09, q = 0.49). In squamous NSCLC, IL-10 at 3 months correlated inversely with survival (ρ = −0.71, 95% CI −0.93 to −0.14, *p* = 0.022, *n* = 10) and IL-2 at 6 months correlated positively (ρ = 0.90, 95% CI 0.09 to 0.99, *p* = 0.037, *n* = 5); however, neither remained significant after BH–FDR (q ≈ 0.17), and the 6-month IL-2 estimate is limited by small sample size. These exploratory correlations are directionally aligned with the time-updated Cox analyses (protective IL-2, adverse IL-10 signals) but should be interpreted cautiously.

The analysis of baseline cytokine levels in the melanoma cohort revealed mixed correlations with treatment efficacy. A strong positive correlation was observed between baseline TNF-α and IL-10 levels (Spearman’s ρ = 0.665, *p* = 0.007), suggesting coordinated regulation of these cytokines. However, neither TNF-α (ρ = 0.154, *p* = 0.584) nor IL-2 (ρ = −0.184, *p* = 0.511) showed significant associations with overall survival. Similarly, baseline IL-10 levels did not correlate significantly with survival (ρ = 0.104, *p* = 0.713). Logically, a very strong positive correlation emerged between survival duration and best treatment response (ρ = 0.898, *p* < 0.001), indicating that clinical response strongly predicts survival. The lack of significant cytokine-outcome correlations may reflect the small sample size, as evidenced by wide confidence intervals for some associations (e.g., TNF-α/IL-10 correlation: 95% CI 0.216–0.882). These results highlight the complex relationship between baseline inflammatory markers and treatment efficacy in melanoma.

### 3.4. Biomarker Dynamics at 3 Months Timepoint—Correlations with Clinical Outcomes and Treatment Efficacy

The NSCLC cohort established a strong positive correlation between overall survival and best treatment response (ρ = 0.770, *p* < 0.001), with a 95% confidence interval of 0.605–0.871, rationally indicating patients achieving better therapeutic responses experienced prolonged survival. Among cytokines, IL-10 demonstrates clinically relevant associations, showing moderate negative correlations with both survival (ρ = −0.510, *p* = 0.022, 95% CI −0.78–−0.07) and treatment response (ρ = −0.646, *p* = 0.002, 95% CI −0.85–−0.27). This dual affiliation states that IL-10 overexpression at the 3-month timeframe may serve as a biomarker for unfavorable prognosis and suboptimal treatment efficacy.

TNF-α and IL-2 show divergent patterns, with TNF-α exhibiting a borderline negative correlation to treatment response (ρ = −0.444, *p* = 0.050) but no significant survival association (ρ = 0.056, *p* = 0.816). IL-2 demonstrates no statistically significant relationships with either survival (ρ = 0.062, *p* = 0.796) or treatment response (ρ = −0.393, *p* = 0.086). The cytokine network analysis reveals strong intra-system coordination, with TNF-α correlating positively with both IL-2 (ρ = 0.785, *p* < 0.001) and IL-10 (ρ = 0.600, *p* = 0.005), while IL-2 and IL-10 show moderate positive association (ρ = 0.486, *p* = 0.030)—Figure 4. These interrelationships suggest potential shared regulatory mechanisms or feedback loops within the tumor microenvironment.

Similarly, in the melanoma cohort, cytokine levels measured at the three-month follow-up interval exhibited statistically significant correlations with both overall survival rates and clinical treatment outcomes. Both survival time and treatment response showed significant inverse relationships with TNF-α levels at 3 months (ρ = −0.821, *p* = 0.023 and ρ = −0.896, *p* = 0.006, respectively). Notably, IL-10 exhibited strong positive correlations with both TNF-α (ρ = 0.929, *p* = 0.003) and IL-2 (ρ = 0.786, *p* = 0.036). The TNF-α/IL-2 relationship showed borderline significance (ρ = 0.714, *p* = 0.071). Confidence intervals revealed particularly wide ranges for survival-TNF-α (95% CI −0.974 to −0.151) and IL-10-IL-2 associations (95% CI 0.051–0.969), suggesting substantial variability in these relationships—Figure 5. These results indicate that elevated TNF-α levels at 3 months may predict poorer clinical outcomes, while IL-10 shows strong covariation with both TNF-α and IL-2. The small sample size (*n* = 7 for cytokine analyses) necessitates cautious interpretation, particularly regarding the IL-2/TNF-α relationship where statistical significance was marginal, analysis being more of an exploratory nature rather than a core claim.

## 4. Discussion

Despite a limited number of patients, this study provides preliminary insight into the potential predictive role of serum cytokines, TNF-α, IL-2 and IL-10 in patients diagnosed with NSCLC and melanoma undergoing anti-PD-1 immunotherapy with Nivolumab. The findings are contextualized within the broader landscape of immunotherapy biomarker research, emphasizing the clinical need for dynamic, cost efficient and reliable predictors of both efficacy and toxicity in immune checkpoint inhibitor treatment [23,24,25,26].

This study’s novelty lies in its prospective longitudinal design, differentiating cytokine trends by cancer type (NSCLC vs. melanoma), and correlating biomarker changes at distinct timepoints with clinical outcomes. Additionally, IL-10 dynamics as a potential early signal of poor response in NSCLC, a finding that to our knowledge has not previously been reported with comparable granularity in a real-world setting.

### 4.1. Cytokine Dynamics

The longitudinal monitoring of cytokine levels revealed distinct temporal and disease-specific patterns. As highlighted in this study, both NSCLC and melanoma patients exhibited complex TNF-α trajectories. However, melanoma cohort displayed a more consistent increase over time, which may suggest that pro-inflammatory cytokine activation is more pronounced, though this observation requires validation and may contribute to, but does not fully explain, its superior response to immunotherapy compared to NSCLC [27,28,29]. In the present study, a subset of NSCLC patients experienced TNF-α level fluctuations, this being already reported and associated with transient immune activation and stromal resistance mechanisms [30,31,32]. Notably, TNF-α expression in NSCLC has been linked to both favorable and unfavorable prognosis depending on its anatomical localization, with islet expression correlating with improved survival and stromal expression with reduced survival [30]. Within our investigation, most of the subjects who experienced TNF-α fluctuations were patients with disease progression, consistent with prior reports that dynamic TNF-α signaling may reflect both anti-tumor immunity and resistance mechanisms. [33,34].

NSCLC patients displayed heterogeneous IL-2 kinetics, with some showing high baseline levels that declined over time and others maintaining low levels throughout coinciding with existing studies that report an imbalance of the IL-2/IL-2 receptor system [35] in advanced NSCLC and its association with disease activity and progression [36,37,38]. In contrast, melanoma patients demonstrated marked and sustained IL-2 elevations, particularly by six months, which may be compatible with more robust T-cell activation, though our small sample size limits firm conclusions. These findings are consistent with the higher immunogenicity [39] and response rates of melanoma to PD-1 blockade [40,41,42,43]. The role of IL-2 in expanding tumor-infiltrating lymphocytes and supporting durable responses is well documented, with high-dose IL-2 therapy historically approved for metastatic melanoma and associated with improved outcomes in selected patients [36,44]. The variable IL-2 kinetics, elevated in some NSCLC cases at baseline but sustained in melanoma, likely reflect differing T-cell functional states. In NSCLC, down trending IL-2 may signal T-cell dysfunction or regulatory T-cell skewing, suggesting an immunosuppressive milieu or exhaustion phenotype, whilst in melanoma, IL-2 elevations may indicate heightened CD8+ expansion, consistent with its known immunogenicity.

Regarding IL-10, in our research, NSCLC patients generally maintained low IL-10 levels, with rare exceptions. This is compatible with broad literature where NSCLC patients generally exhibit low IL-10 expression, particularly in squamous cell carcinoma and adenocarcinoma, as shown by mRNA downregulation in tumor regions [45,46]. IL-10 in NSCLC has been linked to tumor tolerance, larger tumor size, and regulatory T-cell infiltration, but its overall expression remains suppressed compared to surrounding tissue [46]. Melanoma patients showed more variable IL-10 trends, which may indicate heterogeneous regulatory immune responses, including possible immune evasion via T-cell apoptosis [47,48] or downregulation of antigen presentation on dendritic cells [49,50,51]. IL-10 can also suppress angiogenesis by inhibiting macrophage-derived VEGF, highlighting its dual role in both immune suppression and anti-angiogenic activity [52,53]. Notably, a transient spike in IL-10 in one NSCLC patient did not correlate with immune-related adverse events (irAEs), suggesting that IL-10 fluctuations may reflect regulatory feedback rather than toxicity alone [54].

Transforming and standardizing cytokines stabilized estimation and put markers on a comparable scale, which likely explains why the time-updated joint model revealed significant associations that were not apparent in static/untransformed analyses. The directionally opposite effects (IL-2 protective, TNF-α adverse) are biologically plausible for on-treatment immune dynamics.

### 4.2. Timeline Dynamics of Biomarkers: Associations with Clinical Endpoints and Therapeutic Efficacy

Our statistical analysis employing linear mixed models and Spearman correlation tests uncovered several key connections. For the NSCLC patients, baseline TNF-α levels correlated positively with IL-2 and IL-10, indicating coordinated inflammatory signaling. Importantly, higher baseline TNF-α was associated with poorer survival and treatment response, suggesting that TNF-α may function as a negative prognostic biomarker, pending confirmation in larger cohorts. These findings are congruent with previous studies where TNF-α was identified as a dual mediator of immune regulation in cancer, where persistent systemic elevation may drive T cell exhaustion and limit therapeutic efficacy of PD-1 blockade [29,55,56].

On the contrary, baseline IL-2 and IL-10 levels showed no significant associations with clinical outcomes in NSCLC. The lack of predictive value for IL-2 might reflect its pleiotropic function, balancing effector T cell activation and regulatory T cell expansion [57,58], whilst IL-10 is seen as an immunosuppressive cytokine that may hold prognostic relevance only under dynamic changes rather than at static baseline concentrations [59,60,61].

In the melanoma cohort, baseline cytokine correlations also pointed to shared regulatory mechanisms, particularly between TNF-α and IL-10. However, none of the cytokines at this time point were significantly associated with survival or treatment response. This contrasts with literature suggesting baseline inflammatory signatures may predict immunotherapy responsiveness in melanoma [15,62,63]. Importantly, the limited sample size of 7 patients likely constrained statistical power, hence limiting the interpretability of these results.

Longitudinal cytokine profiling at three months demonstrated more striking connections regarding clinical outcomes in both cohorts. In NSCLC cohort, IL-10 emerged as a potential biomarker, showing exploratory signals of negative correlations with survival and treatment efficacy. This suggests that early IL-10 elevation may predict a tumor-driven immunosuppressive feedback mechanism, which diminishes T-cell mediated anti-tumor immunity even during PD-1 blockade [64,65]. Similar findings were reported before where IL-10 elevation was associated with inferior survival in stage IV melanoma patients, reinforcing the possibility that IL-10 may hold prognostic value as an early marker [66].

TNF-α levels at three months were only marginally associated with response and not with survival, while intra-cytokine correlations remained strong. These associations suggest an orchestrated cytokine network possibly shaped by the tumor microenvironment or treatment immune modulation [67]. The persistence of such inflammatory signaling has been previously linked to reduced treatment benefit and immune related adverse events [68].

In the melanoma cohort, while baseline cytokine levels did not significantly predict survival, 3-month TNF-α levels were inversely associated with both survival and treatment response. This timeline shift in prognostic relevance highlights the need of dynamic immune profiling during immune checkpoint inhibitors. Furthermore, IL-10 exhibited strong positive correlations with both TNF-α and IL-2, aligning with proposed models of compensatory immune regulation, though our results remain hypothesis-generating [69,70].

Interestingly, the observed strong inter-cytokine correlations, particularly between TNF-α and IL-10, in both cancer types may reflect a conserved immunoregulatory axis, but this interpretation should be viewed as exploratory [71]. Although IL-2 did not independently predict outcomes, its tight correlation with other cytokines suggests its role may be contextual and network-dependent rather than linear [72]. Prior studies have suggested that IL-2’s impact on therapeutic outcomes may vary depending on the cytokine milieu and the timing of expression [44,73].

The study’s real-world, single-center design enhances the relevance of its findings but also introduces limitations, notably the high attrition rate for longitudinal sampling (53.4% at three months, 74.1% at six months). This reflects the practical challenges of biomarker monitoring in advanced cancer populations, where rapid clinical deterioration is common. Although no imputation was performed, comparative analyses at baseline suggest minimal bias. Nonetheless, the possibility of unmeasured confounding cannot be excluded. The use of robust statistical methods, including linear mixed-effects modeling and non-parametric correlation analyses, strengthens the validity of the observed associations despite these limitations. Importantly, the effect of prior systemic treatments or therapy-free intervals (“paused therapy”) on cytokine dynamics was not explored in this study. Given the immunomodulatory effects of prior chemotherapy or targeted agents, future research should aim to stratify patients by treatment history, which may help clarify the immunological effects of ICIs more precisely. Also, we cannot determine whether the observed associations (e.g., protective IL-2 and adverse TNF-α signals) persist or attenuate with prolonged therapy. Attrition is likely informative, so patients remaining in follow-up may differ biologically from those who discontinued earlier. Although time-updated Cox models align risk with the most recent cytokine values, they do not capture trajectories beyond 6 months. Studies with extended serial sampling and, ideally, joint longitudinal–survival modeling are needed to assess durability.

Future studies should validate these findings in larger, prospective cohorts, ideally incorporating both systemic and tissue-level immune profiling since this study focused solely on serum cytokine dynamics. Beyond IL-2, IL-10, and TNF-α, incorporating a broader panel of soluble biomarkers, could refine biological interpretation and improve patient selection for anti-PD-1 therapy [74]. Future work should integrate tumor microenvironment profiling, such as tumor-infiltrating lymphocytes (TILs), PD-L1 expression, and spatial transcriptomics, to contextualize systemic immune signatures and validate serum biomarkers with intratumoral correlates. Additionally, multi-dimensional immune monitoring, including T cell phenotyping and gene expression analyses, may enhance our understanding of the interplay between cytokine dynamics and therapeutic efficacy.

## 5. Conclusions

This study provides preliminary evidence that dynamic cytokine monitoring may help refine prognostic assessment in patients receiving immune checkpoint inhibitors. Among the signals observed, elevated IL-10 levels at the 3-month mark in NSCLC patients emerged as the most consistent and clinically plausible finding, being associated with poorer survival and treatment response. This observation should be considered hypothesis-generating and requires validation in larger, independent cohorts.

Other cytokine trajectories, including IL-2 and TNF-α patterns, showed variability across cancer types and timepoints, but given the limited sample size, particularly in melanoma, these results remain exploratory. The observed inter-cytokine correlations suggest that systemic immune regulation may operate through network effects, yet such interpretations should be viewed with caution.

These findings highlight the importance of dynamic, rather than static, immune profiling for predicting therapeutic response and prognosis in immune checkpoint inhibitor-treated patients.

While the overall cohort was limited due to the nature of advanced-stage disease and associated mortality, this reflects real-world clinical attrition and underscores the challenges of maintaining longitudinal biomarker monitoring in such populations.

Robust statistical analyses lend credibility to the results, yet larger, prospective studies are needed for validation. Future research should integrate systemic and tissue-level immune profiling, as well as advanced immunophenotyping, to better elucidate the interplay between cytokine dynamics and immunotherapy outcomes. Ultimately, such efforts may pave the way for more personalized and effective biomarker-driven cancer immunotherapy strategies.

## Figures and Tables

**Figure 1 cimb-47-00746-f001:**
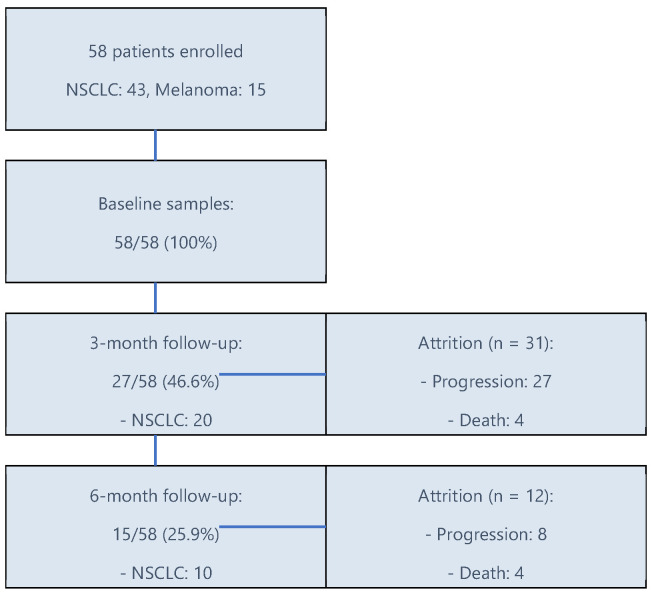
**Flow diagram of patient enrollment, follow-up, and attrition.** A total of 58 patients (NSCLC: 43; melanoma: 15) were treated with nivolumab; baseline samples were available for all, with attrition mainly due to progression and death across follow-up.

**Figure 2 cimb-47-00746-f002:**
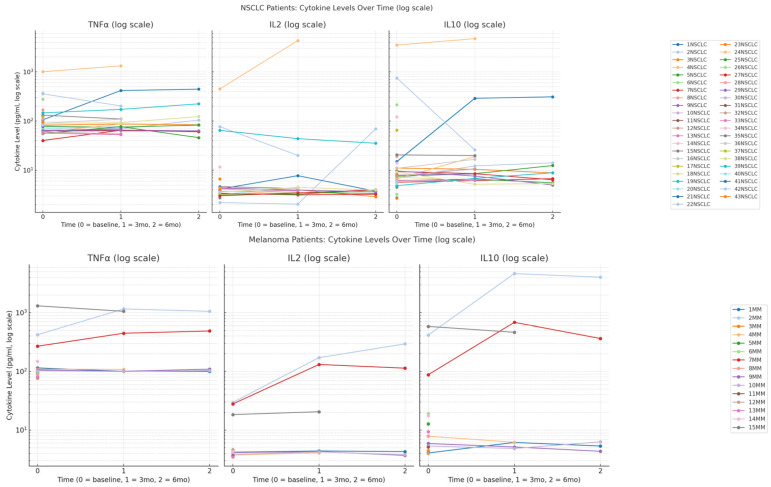
Longitudinal changes in serum cytokine levels (IL-2, IL-10, and TNF-α) in patients with non-small cell lung cancer (NSCLC; top row, *n* = 43 at baseline; *n* = 20 at 3 months; *n* = 10 at 6 months) and melanoma (bottom row, *n* = 15 at baseline; *n* = 7 at 3 months; *n* = 5 at 6 months) undergoing treatment with Nivolumab.

**Figure 3 cimb-47-00746-f003:**
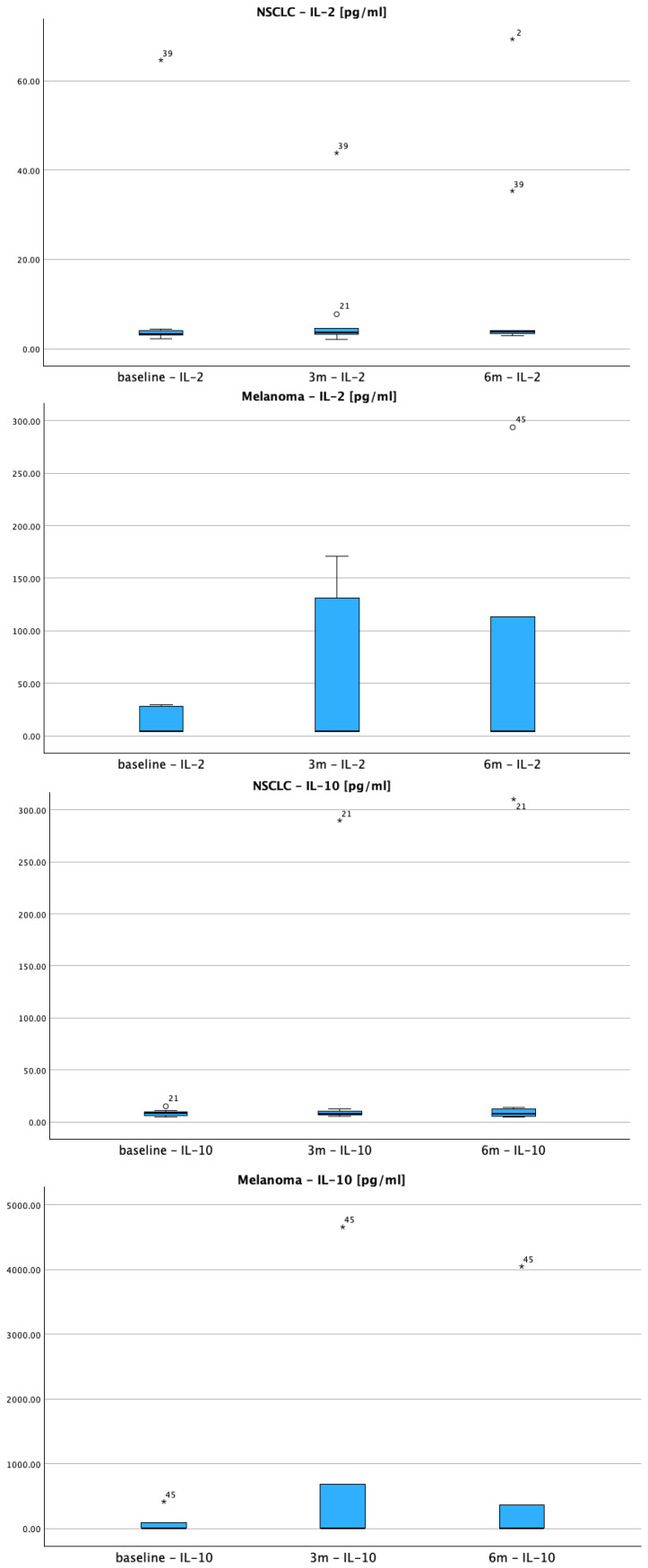
Comparative temporal trends of serum cytokine levels (IL-2, IL-10, and TNF-α) in NSCLC (*n* = 43 at baseline; *n* = 20 at 3 months; *n* = 10 at 6 months) and melanoma (*n* = 15 at baseline; *n* = 7 at 3 months; *n* = 5 at 6 months) cohorts undergoing anti-PD-1 therapy. “*”, “°” represent outliers.

**Figure 4 cimb-47-00746-f004:**
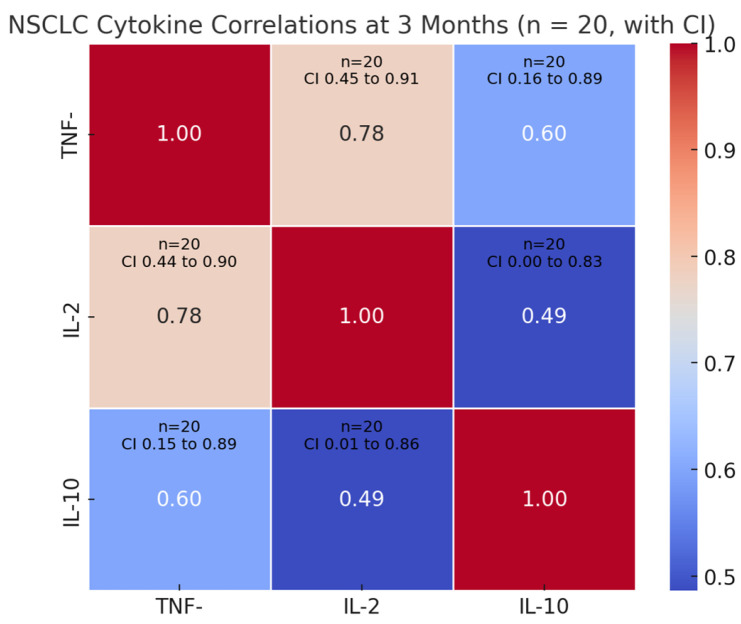
NSCLC cohort—cytokine intercorrelations at 3 months. This heatmap illustrates Spearman’s rank correlation coefficients between serum cytokine levels (TNF-α, IL-2, and IL-10) in NSCLC patients 3 months after initiation of Nivolumab therapy. The strength and direction of correlation are color-coded, ranging from strong negative (blue) to strong positive (red). Numerical values within each cell represent the correlation coefficient (ρ). This visualization reveals strong intra-cytokine coordination, especially between TNF-α and IL-2 (ρ = 0.785) and between TNF-α and IL-10 (ρ = 0.600), suggesting synchronized inflammatory and regulatory signaling.

**Figure 5 cimb-47-00746-f005:**
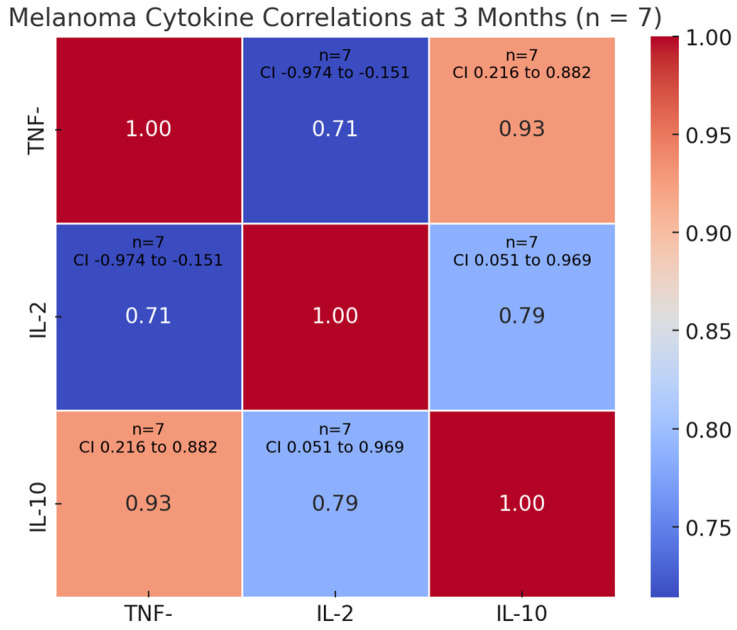
Melanoma cohort—cytokine intercorrelations at 3 months. This heatmap illustrates the Spearman correlation coefficients (ρ) among TNF-α, IL-2, and IL-10 in melanoma patients 3 months following initiation of anti-PD-1 therapy with Nivolumab. Warmer colors represent stronger positive correlations, and cooler tones represent weaker or absent associations. The matrix reveals a robust immune cytokine network, with very strong positive correlation between TNF-α and IL-10 (ρ = 0.929), and strong associations between IL-2 and IL-10 (ρ = 0.786), and TNF-α and IL-2 (ρ = 0.714). These patterns indicate enhanced and coordinated immune activation in melanoma patients under checkpoint blockade therapy.

**Table 1 cimb-47-00746-t001:** Patients’ characteristics.

Characteristics	Value/Distribution
**Number of patients**	58
**Median age (years)**	62.5 (range: 35–83)
**Cancer type**	Lung: 43 (77.6%)
	Melanoma: 15 (22.4%)
**Sex**	Male: 39 (67.2%)
	Female: 19 (32.8%)
**Smokers**	Yes: 41 (70.7%)
	No: 17 (29.3%)
**Use of corticosteroids at baseline**	Yes: 21 (36.2%)
	No: 37 (63.8%)
**Type of NSCLC**	Adenocarcinoma 25 (58.1%)
	Squamous cell carcinoma 18 (41.9%)
**PD-L1**	Positive >1% 22 (50%)
	Negative <1% 22 (50%)
**ECOG**	0–31 (53.45%)
	1–20 (34.48%)
	2–7 (12.07%)
**Median overall survival (months)**	21.1
**Best response**	Progression 48.3% (28), Stable disease 43.1% (25),
	Complete response 6.9% (4), Partial response 1.7% (1)
**Median number of Nivolumab cycles**	17
**Average line of Nivolumab**	2
**Line of Nivolumab**	1–13 (22.4%)
	2–32 (58.6%)
	3–7 (12.1%)
	4–4 (6.9%)
**Subsequent lines of therapy**	50.0% (29/58)
	Most frequent: Docetaxel, Carboplatin + Paclitaxel, Gemcitabine, Dabrafenib + Trametinib
**Previous lines of treatment**	Carboplatin + Pemetrexed 19 (32.76%)
	Carboplatin +Gemcitabine 11 (18.97%)
	Carboplatin + Paclitaxel 5 (8.62%)
	Gemcitabine 3 (5.17%)
	Carboplatin + Navelbine 2 (3.45%)
	Carboplatin 2 (3.45%)
	Dacarbazine 2 (3.45%)
	Dabrafenib + Trametinib 2 (3.45%)
	No previous therapy: 20.69%

## Data Availability

The raw data supporting the conclusions of this article will be made available by the authors on request.

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
