# Peer review of "Serum TNF -α, IL-10 and IL-2 Trajectories and Outcomes in NSCLC and Melanoma Under Anti-PD-1 Therapy: Longitudinal Real-World Evidence from a Single Center"

_cimb, 2025, doi:10.3390/cimb47090746_

Round 1
Reviewer 1 Report (New Reviewer)
Comments and Suggestions for Authors
The manuscript's topic is truly interesting, and the article is well-written and structured. However, some aspects could be improved:
I appreciated that the authors highlighted most of the flaws in their analysis in the "Discussion" section, namely: the lack of correlation with histological tests, the lack of correlation with other cytokines or potential biomarkers, and the differences between melanoma and NSCLC. However, the authors cannot mitigate the shortcomings of their study by hoping for future studies, as some of these findings could be extracted and thus improve the manuscript as a whole.
1) Especially for lung cancer cases, it would be helpful to specify the histotype, also to better identify any differences in immunosensitivity and any differences in previous treatment choices. We know that cytokines vary greatly over time based on various factors, including the type of previous treatment.
2) It would be useful to compare data on histological specimens derived from the same cases to determine whether there were significant differences in TME that could have influenced the study results.
3) It would also be interesting to know if, and how many patients (and for how long) took steroids, as this may also have influenced the results.
4) Given that therapy was prolonged for much longer, it would have been useful to assess whether these results were maintained over time. If this is no longer possible, the authors should specify this in a discussion section as a weakness of the study.
5) Given that the authors decided to test only soluble factors in patients treated with anti-PD1 ICIs, it would be useful to specify why they did not test other factors, such as soluble CD73, which has been shown for several years to correlate with ongoing responses to nivolumab in patients with metastatic melanoma (Morello, 2017).
Author Response
Please find below the response to your comments in the attached letter.

Reviewer 2 Report (New Reviewer)
Comments and Suggestions for Authors
This prospective single-center cohort explores longitudinal serum TNF-α, IL-10 and IL-2 dynamics in stage IV NSCLC and melanoma patients receiving nivolumab, relating cytokine trajectories (baseline, 3 and 6 months) to response and survival. The real-world design and serial sampling are valuable. However, there are important issues that limit interpretability and reproducibility. I recommend major revision to address the points below.
Title:
- Grammar: current phrasing “predictive biomarkers to immunotherapy … real-world data” is awkward and “data” is plural. Consider:
“Longitudinal Serum TNF-α, IL-10, and IL-2 as Predictors of Anti-PD-1 Outcomes in NSCLC and Melanoma: A Single-Center Real-World Cohort.”
- Ensure the scope (NSCLC and melanoma; nivolumab only) and the longitudinal nature appear in the title.
Abstract:
- Strengthen quantitative reporting: state sample sizes per tumor type and per timepoint (baseline/3/6 months), and give key effect sizes with CIs (e.g., IL-10 at 3 months vs survival in NSCLC: ρ and 95% CI). Avoid broad phrasing like “strong correlations” without numbers.
- Temper causal wording (“predictive”) unless predictions were prospectively specified and validated; “associations” may be more appropriate given exploratory analyses and small n.
- Clarify that baseline markers had limited utility particularly due to small melanoma n, and that the most informative signal was 3-month IL-10 in NSCLC.
Introduction:
- The background is generally comprehensive but occasionally drifts to topics not used here (e.g., multiplex/ML pipelines and oncolytic peptides) creating scope creep. Please streamline to cytokines measurable by ELISA and rationale for TNF-α/IL-10/IL-2 specifically, with a sharper statement of the a priori primary endpoint (if any).
- Articulate the biological hypotheses per tumor type (NSCLC vs melanoma) and pre-specify which cytokine/timepoint associations were primary vs exploratory.
Methods:
- The manuscript calls this a “prospective interventional” study, but nivolumab appears to have been administered as standard care in a single center. Please reclassify as a prospective observational cohort and clarify any protocolized interventions beyond routine care. Include a STROBE diagram for enrollment, follow-up, and reasons for attrition at 3 and 6 months.
Cytokine assays
- ELISA kits, CVs, and LODs are reported—good. Add the handling of values below LOD (e.g., half LOD or censoring approach). State preanalytical controls (fasting status, time-of-day), storage conditions, and batch effects (were all timepoints for a patient run on the same plate?).
Clinical covariates
- Please report and consider adjusting for PD-L1 status, smoking, ECOG PS, line of therapy, steroids/antibiotics, BRAF (melanoma), and key labs (CRP, albumin). These are potential confounders for both cytokines and outcomes. Currently, the models appear unadjusted.
Statistical analysis
- You state linear mixed-effects models (random intercepts, AR(1)), and numerous Spearman correlations at multiple timepoints and within two tumor types. Given the volume of tests, control type-I error (e.g., FDR). Pre-define primaries (e.g., NSCLC 3-month IL-10 vs OS) and treat others as exploratory.
- For LMMs, report full model specification: fixed effects (time, tumor type, interaction), covariance structure selection strategy, estimation (REML/ML), and diagnostics (normality of residuals). Provide model-based estimates with 95% CIs and exact p-values.
- Missingness: attrition is substantial (≈53% at 3 months; ≈74% at 6 months). MAR is assumed but not verifiable; given that drop-out is likely informative (progression/death), consider joint longitudinal-survival models, pattern-mixture/sensitivity analyses, or at minimum show baseline comparisons of those with vs without follow-up cytokines (table in Supplement). Clarify how deaths before 3/6 months were handled in analyses.
- Correlation heatmaps: include n for each matrix, and show CIs alongside ρ. For very small melanoma n at 3 months, caution against over-interpretation.
Results:
Cohort description
- Table 1 is useful; please add PD-L1, ECOG, line of therapy distribution, prior systemic therapies, and adverse events (especially immune-related). Report exact numbers behind the percentages of best response categories.
Longitudinal cytokines
- When stating “trend toward significance” (e.g., TNF-α time effect p≈0.054), either avoid “trend” language or define it a priori; better to report the estimate and CI and refrain from quasi-significance framing.
Associations with outcomes
- The core claims (e.g., NSCLC 3-month IL-10 inversely associated with OS/response) are interesting, but provide effect sizes with CIs and adjusted analyses. For melanoma, emphasize the very small n at 3 months (you note n≈7), and present sensitivity analyses or move such results to Supplement emphasizing exploratory nature.
Figures & Tables:
- Figure 1–2 (trajectories/boxplots): add sample sizes per panel/timepoint.
- Figure 3–4 (heatmaps): add n and 95% CIs for each ρ.
Discussion:
- The discussion usefully contrasts NSCLC vs melanoma but occasionally over-extends mechanistic inference. Please temper claims, especially where sample sizes are small or p-values are marginal.
- Align discussion strictly with assayed cytokines and measured endpoints; remove or move to perspective any content on oncolytic peptides/AI pipelines not used here.
- Expand limitations: single center; substantial attrition with likely informative dropout; lack of PD-L1 and other covariates in models; multiple testing; small melanoma follow-up n; absence of external validation.
- Propose a clear next-step validation plan (pre-registered endpoints, independent cohort, joint modeling, multiplex cytokine panel).
Conclusions:
- Center the conclusion on the most robust and clinically plausible finding (3-month IL-10 in NSCLC), explicitly labeling it as hypothesis-generating pending validation.
Author Response
Please find the attached letter.

Round 2
Reviewer 1 Report (New Reviewer)
Comments and Suggestions for Authors
The authors have revised the manuscript in accordance with the reviewers' suggestions, making it worthy of publication. Congratulations to the authors!
Reviewer 2 Report (New Reviewer)
Comments and Suggestions for Authors
The authors responded appropriately to the reviewer's requests/corrections. I recommend accepting the manuscript.
This manuscript is a resubmission of an earlier submission. The following is a list of the peer review reports and author responses from that submission.
Round 1
Reviewer 1 Report
Comments and Suggestions for Authors
This single-center prospective study investigated TNF-α, IL-2, and IL-10 as dynamic biomarkers in 58 stage IV NSCLC (n=45) and melanoma (n=13) patients treated with Nivolumab. Longitudinal serum analysis (baseline/3/6 months) revealed melanoma-specific sustained IL-2/TNF-α increases versus heterogeneous patterns in NSCLC. Elevated IL-10 at 3 months correlated with poorer NSCLC survival. Strong TNF-α/IL-10 inter-correlations suggested shared immunoregulatory pathways. Collectively, this manuscript is recommended to be accepted after some minor revisions.
- Recently, oncolytic peptides possessing synergistic oncolytic-immunotherapy effect have emerged as one of the main subsets of immunotherapy agents, and multiple oncolytic peptides have entered clinical trials for immunotherapy. To help readers and potential users, the representative work on the development and immunotherapy application of oncolytic peptides should be cited (suggest, J. Med. Chem., 2024, 67, 3885. Acta Pharmacol. Sin., 2023, 44, 201. Bioorg. Chem., 2023, 138, 106674.).
- Page 5, Figure 2. The font in Figure 2 is too small and has low definition. Could the author please adjust the font size and improve the clarity in Figure 2?
- Page 4, Figure 1. Figure 1 shows that patients with melanoma who received nivolumab experienced an increase in IL-2 levels. Could the authors please indicate which immune cells are likely responsible for the increase and whether it is related to the degree of immune cell activation?
- Page 6, line 168. The author analyzed 43 patients with non-small cell lung cancer, but Table 1 includes 45 patients with lung cancer. Please check.
This single-center prospective study investigated TNF-α, IL-2, and IL-10 as dynamic biomarkers in 58 stage IV NSCLC (n=45) and melanoma (n=13) patients treated with Nivolumab. Longitudinal serum analysis (baseline/3/6 months) revealed melanoma-specific sustained IL-2/TNF-α increases versus heterogeneous patterns in NSCLC. Elevated IL-10 at 3 months correlated with poorer NSCLC survival. Strong TNF-α/IL-10 inter-correlations suggested shared immunoregulatory pathways. Collectively, this manuscript is recommended to be accepted after some minor revisions.
- Recently, oncolytic peptides possessing synergistic oncolytic-immunotherapy effect have emerged as one of the main subsets of immunotherapy agents, and multiple oncolytic peptides have entered clinical trials for immunotherapy. To help readers and potential users, the representative work on the development and immunotherapy application of oncolytic peptides should be cited (suggest, J. Med. Chem., 2024, 67, 3885. Acta Pharmacol. Sin., 2023, 44, 201. Bioorg. Chem., 2023, 138, 106674.).
- Page 5, Figure 2. The font in Figure 2 is too small and has low definition. Could the author please adjust the font size and improve the clarity in Figure 2?
- Page 4, Figure 1. Figure 1 shows that patients with melanoma who received nivolumab experienced an increase in IL-2 levels. Could the authors please indicate which immune cells are likely responsible for the increase and whether it is related to the degree of immune cell activation?
- Page 6, line 168. The author analyzed 43 patients with non-small cell lung cancer, but Table 1 includes 45 patients with lung cancer. Please check.
Author Response
Please find attached below the response letter to your comments.

Reviewer 2 Report
Comments and Suggestions for Authors
Manuscript Overview
This manuscript presents a prospective, longitudinal analysis of serum cytokines TNF-α, IL-2, and IL-10 in patients with advanced NSCLC and melanoma undergoing anti-PD-1 immunotherapy (Nivolumab). It explores temporal cytokine dynamics and their associations with clinical outcomes, aiming to identify dynamic immune biomarkers predictive of treatment response and survival.
General Comments
The study addresses an important clinical problem: identifying biomarkers for predicting and monitoring immunotherapy efficacy. Longitudinal cytokine profiling is a strength, capturing dynamic immune changes rather than relying on single time points. The combination of NSCLC and melanoma cohorts allows comparison of disease-specific immune responses. Statistical approaches, including linear mixed models and Spearman correlations, are appropriate for longitudinal data. The manuscript is generally well-written and contextualizes findings within current literature. However, there are limitations should be edited as the following:
Special Comments
- Can you clarify the sample size used for cytokine analyses? How do you address the limited statistical power given only 7 patients were analyzed longitudinally?
- How did you handle missing data from the high attrition rates at 3 and 6 months? Were any imputation or sensitivity analyses performed?
- What was the rationale for combining NSCLC and melanoma cohorts? Could you provide separate analyses or justify pooling these distinct tumor types?
- Please provide more details on the ELISA assay validation, including reproducibility, sensitivity, and quality control measures.
- Have you considered integrating tumor microenvironment immune profiling (e.g., TILs, cytokine expression) to complement serum cytokine data?
- The role of IL-2 appears inconsistent; could you elaborate on its kinetics and possible functional implications in your cohort?
- Can you provide graphical representations of cytokine correlations and dynamics, such as heatmaps or network diagrams?
- How were clinical endpoints like treatment response and survival defined and assessed?
Author Response
Please do find below the letter containing the responses to your comments.
Reviewer 3 Report
Comments and Suggestions for Authors
Dear authors, please find advanced summary and comments attached as pdf.

Author Response
We do thank you for your time, please do find below our letter.

Reviewer 4 Report
Comments and Suggestions for Authors
The study was intended to identify the prognostic role of TNF-alpha, IL-10 and IL-2 levels in relation to the efficacy of nivolumab therapy in patients with lung cancer and melanoma. However, the discussion of this challenge is mainly based on literature data speculations rather than on actual data obtained.
- The specific features of the research is not identified. How does it differ, original or novel compared to the reports mentioned in the literature review?
- Efficacy criteria are not formulated. Data on the dynamics of the disease and outcome for each patient are not presented.
- The scattered data are presented from random 3-8 patients from the group and were not studied in the patients declared in the groups (n=45) and (n=13) that do not allow for correct comparison of the groups, assessment of trends and the effect of therapy.
- It remains unclear whether the administration of Nivolumab was the only therapy for these patients that had an impact on the levels of these biomarkers?
The effect of other lines of therapy (paused therapy) on changes in biomarker levels is not investigated or discussed in the paper.
Author Response
Responses have been incorporated in the letter below as per recommendations.
Round 2
Reviewer 2 Report
Comments and Suggestions for Authors
I endorse the publication.
Author Response
Please do find below our response.

Reviewer 4 Report
Comments and Suggestions for Authors
The rearrangement of the text and the additions made the manuscript more organized and clearer. Unfortunately, the actual data presented in the work are very limited and contradictory to convince that TNF-α, IL-10 and IL-2 can serve as reliable prognostic biomarkers of the effectiveness of immunotherapy.
In my opinion, this study requires additional collection and analysis of material in order to present more substantiated data.
Fig 1 demonstrates individual cytokine trajectories from 5-8 NSCLC and from 3-4 melanoma patients. The number of patients examined is usually indicated. As can be seen from the figure, each parameter was studied in a different number of patients, which is not equivalent when comparing markers and cannot be explained by premature attrition of patients.
Why are data from other patients (“cytokine data were available at 3 and 6 months for 27 and 15 patients”) not shown? The interpretation of the results does not correspond to reality. L231 "... stronger activation of proinflammatory cytokines (IL-2, TNF-α)" is not supported by the presented data, due to their inconsistency.
Fig 2. Designations are unreadable.
Author Response
Please do find below the letter concerned.
